# Mitigation of Cardiovascular Disease and Toxicity through NRF2 Signalling

**DOI:** 10.3390/ijms24076723

**Published:** 2023-04-04

**Authors:** James A. Roberts, Richard D. Rainbow, Parveen Sharma

**Affiliations:** 1Department of Cardiovascular and Metabolic Medicine, Institute of Life Course and Medical Sciences, University of Liverpool, Liverpool L7 8TX, UK; james.roberts@liverpool.ac.uk (J.A.R.); richard.rainbow@liverpool.ac.uk (R.D.R.); 2Liverpool Centre for Cardiovascular Science, Liverpool L7 8TX, UK

**Keywords:** NRF2, cardiotoxicity, cardiovascular disease, doxorubicin, ageing

## Abstract

Cardiovascular toxicity and diseases are phenomena that have a vastly detrimental impact on morbidity and mortality. The pathophysiology driving the development of these conditions is multifactorial but commonly includes the perturbance of reactive oxygen species (ROS) signalling, iron homeostasis and mitochondrial bioenergetics. The transcription factor nuclear factor erythroid 2 (NFE2)-related factor 2 (NRF2), a master regulator of cytoprotective responses, drives the expression of genes that provide resistance to oxidative, electrophilic and xenobiotic stresses. Recent research has suggested that stimulation of the NRF2 signalling pathway can alleviate cardiotoxicity and hallmarks of cardiovascular disease progression. However, dysregulation of NRF2 dynamic responses can be severely impacted by ageing processes and off-target toxicity from clinical medicines including anthracycline chemotherapeutics, rendering cells of the cardiovascular system susceptible to toxicity and subsequent tissue dysfunction. This review addresses the current understanding of NRF2 mechanisms under homeostatic and cardiovascular pathophysiological conditions within the context of wider implications for this diverse transcription factor.

## 1. Introduction

The maintenance of a healthy cardiovascular system is critical to the survival and wellbeing afforded to mammalian life. Unfortunately, the development of cardiac disorders and cardiovascular diseases (CVD) is a major factor determining human morbidity and mortality, with incidence rates significantly increasing worldwide. A comprehensive study of CVD trends across a 25-year period noted that over 400 million people globally suffered from CVD in 2015 with almost 18 million deaths, an increase of approximately five million deaths since 1990, which constituted one third of all deaths that year [1]. Evidently, despite the vast focus from health organisations worldwide, the discovery of detection and intervention strategies to mitigate the early onset and progression of CVD, as well as treatment opportunities for patients with clinically recognised comorbidities, is still of great importance.

The development of CVD and other cardiac disorders is a highly complex biological phenomenon and is difficult to dissect into individual pathophysiological mechanisms. Cells within the cardiovascular system are repeatedly exposed to potentially toxic substances, whether from external sources (such as pharmaceutical drugs) or endogenous signalling molecules that maintain homeostasis such as reactive oxygen species (ROS), but in high abundance can act as toxic stimuli [2]. To combat this, there are many intracellular physiological mechanisms that have adapted to preserve a healthy cardiovascular status and many pharmaceutical agents in the clinic today that target these processes for therapeutic gain: statins for atherosclerosis [3], angiotensin-converting enzyme inhibitors for hypertension [4] and metformin for cardiovascular complications in diabetes [5]. Typically, however, these drugs display efficacy when treating the symptoms of CVD and will only be prescribed when the condition presents clinically, by which time many of the detrimental and often irreversible damage has occurred. Perhaps, alternatively, there are strategies by which physiological pathways can be harnessed in much earlier stages of CVD to alleviate the toxicity caused by the initial stimuli and thus protect healthy cells and tissues from the development of diseases. It must be noted that this would require improved detection methodologies and a more prophylactic approach to medicine.

One such mechanism that has gained a large research focus within the past two decades is that of the transcription factor nuclear factor erythroid 2 (NFE2)-related factor 2 (NRF2) due to the clear indication that stimulation of the NRF2 pathways could prove cytoprotective. Thus, the cardioprotective attributes of NRF2 signalling are being explored. Pharmacological NRF2-activators have routinely proven to reverse disease-like phenotypes in models of almost all aspects of cardiovascular disorders [6] including heart failure [7,8], myocardial infarction [9], cardiomyopathies [8,10], atherosclerosis [11] and fibrosis [12,13]. Importantly, however, NRF2 activation as a cardioprotective mechanism has yet to be translated to patients although, given the rapidly expanding knowledge base regarding this topic, this could be expected within the near future.

Numerous pharmacological modulators of NRF2 have undergone clinical trials across a range of diseases such as nephropathy, myopathies, inflammation, malignancies and respiratory diseases [14]. Dimethyl fumarate, an electrophilic compound that promotes the stabilisation and nuclear translocation of NRF2, has FDA-approval in multiple sclerosis and psoriasis [14,15]. In a phase II clinical trial for type 2 diabetes mellitus (T2DM) patients with chronic kidney disease, a C-28 methyl ester of 2-cyano-3,12-dioxoolean-1,9-dien-28-oic acid-methyl ester (CDDO-Me), another electrophilic NRF2 activator, significantly improved glomerular filtration rates for up to one year beyond treatment [16]. If the aforementioned links between NRF2 and alleviation of cardiac toxicity or disease can be established, it is conceivable that many of these activators could be repurposed for cardiovascular efficacy.

This review will summarise the current understanding of cellular mechanisms driving the development of cardiac disorders and CVD through which manipulation of NRF2 signalling pathways may provide physiological resistance. Most critically, the contribution of NRF2 to preventing initial pathophysiological changes to abrogate damaging events that precede CVD will be discussed. Please note that, for consistency, this review will refer to the protein as NRF2 regardless of the human or animal model in question.

## 2. NRF2 Structure and Regulation

### 2.1. NRF2 Discovery

NRF2 was initially discovered from the Kan research group during experiments measuring β-globin gene regulation and is encoded by the *NFE2L2* gene [17]. Subsequently, in 1997, Itoh et al. characterised NRF2 as an antioxidant-response protein that could induce phase II detoxification enzymes as its homozygous deletion in mice considerably reduced the expression of NAD(P)H quinone oxidoreductase (NQO1) protein [18]. Similarly, NRF2-deficient mice have shown a reduced capacity for the coordination of an antioxidant response upon exposure to several pro-oxidative chemicals [18]. Since then, it has been widely accepted that NRF2 is a master regulator of dynamic antioxidant responses. However, the functions of NRF2 are much more diverse than this. In fact, NRF2 directly targets approximately 1% of the human genome, driving the transcriptional expression of >240 genes with distinct biological functions [19,20,21] including bioenergetics, metabolism of carbohydrates, lipids, iron and upregulation of other transcription factors [22].

### 2.2. NFE2L Family Members

NRF2 is a cap n’ collar (CNC) basic leucine zipper (bZIP) transcription factor. This subfamily of proteins also comprises NFE2, NRF1 and NRF3 [23]. While all three NRF protein family members have shown cytoprotective phenotypes, NRF2 has been most characterised and has undergone the majority of NRF-related scientific focus and, as such, the functional understanding regarding NRF1 and NRF3 has been less well-defined [24]. There is growing evidence supporting the rationale that NRF1 may also play a critical role in normal physiological functions, which may include mediating proteasomal responses, chaperone transcription and protein quality control mechanisms [24,25]. NRF3 is the least well-researched of this family, and many of its physiological functions remain elusive, largely driven by the apparently ineffective phenotypic change upon deletion of the *NFE2L3* gene in vivo despite the severe sensitivity to oxidative stress of both NRF1 and NRF2 knockout [23,26]. However, it is suggested that NRF3 may contribute to the regulation of cell growth and therefore is attracting attention within the field of cancer research [27,28]. There is emerging evidence that NRF1 has an important role within the context of cardiovascular physiology, via redox homeostasis, cytoprotection and neonatal heart regeneration [29]. In the coming years, one may expect the field of NRF1 transcriptomics and the alleviation of cardiovascular diseases or cardiotoxicity to expand. However, due to the significant evidence linking NRF2 to these functional properties to date and the more advanced nature of clinical NRF2 regulatory pharmaceutical agents, this protein will provide the focus for this review.

### 2.3. NRF2 Structure

Human NRF2 is a 66 kDa protein that consists of 605 amino acids and seven highly conserved domains termed NRF2-ECH homology (Neh) regions that each serve unique functions (Figure 1) [30]. The Neh1 domain contains the CNC-bZIP binding sequence through which NRF2 heterodimerises with small musculoaponeurotic fibrosarcoma (sMAF) proteins, required for DNA binding and thus downstream transcriptional activation [31]. Neh2 contains two degron motifs, DLG and ETGE, each of which can interact with one molecule from a KEAP1 homodimer to trigger the ubiquitination and canonical proteasomal degradation of NRF2 and thus repress its activity (Figure 1) [32], as will be discussed in more detail later in this review. Binding between NRF2 and KEAP1 has been described as a “hinge and latch” mechanism whereby stronger binding can be measured at ETGE by approximately 200-fold in comparison to the DLG motif [33]. As such, dissociation of NRF2-KEAP1 binding is more typically observed at DLG rather than ETGE [34,35,36]. Neh3 is a transactivation domain that has received less research focus than several other regions, but is believed to partially modulate transcriptional activation, perhaps via direct interactions with chromodomain ATPase/helicase DNA-binding protein 6 (CHD6) [37,38] since its deletion perturbs ARE-harbouring target gene activation. Similarly, Neh4 and Neh5 are also transactivation domains that exacerbate NRF2 activity upon the binding of CREB-binding protein (CBP) which, in turn, acetylates NRF2 [36,38,39]. Neh6 is an NRF2 regulatory domain similarly to Neh2; however, it mediates a KEAP1-independent mechanism [38,40]. Instead, two degron motifs (DSGIS and DSAPGS) confer binding to a β-transducin repeat-containing protein (β-TrCP) which, as with KEAP1, promotes the ubiquitination and subsequent proteasomal degradation of NRF2 (Figure 1) [41]. Notably, the DSGIS site can be phosphorylated by glycogen synthase kinase (GSK)3β, forming a higher-affinity phosphodegron binding site for β-TrCP [40]. This is not observed for DSAPGS, which perhaps indicates a potential hinge-and-latch mechanism in line with NRF2-KEAP1 interactions at Neh2 [40]. Finally, Neh7 harbours residues for the binding of retinoic X receptor alpha (RXRα) (Figure 1), a known nuclear repressor of NRF2 transcriptional activity [36] as measured through transient knockdown and overexpression studies [42], and thus can be categorised as another transactivation domain [38,42]. It must be noted, however, that the nomenclature of these Neh domains does not follow the N-terminal to C-terminal numerical sequence, but has been retained from historical discoveries [32,38].

### 2.4. Canonical Regulation of NRF2

Under basal conditions, intracellular NRF2 activity can be modulated via the control of transcription and translation but, more prominently, post-translational modifications and degradation pathways [43]. The dominant NRF2 regulatory mechanism is KEAP1-mediated proteasomal degradation. KEAP1 homodimers bind to the aforementioned Neh2 domain of NRF2, but also serve as an adaptor protein for CULLIN3 (CUL3) and RING box protein (RBX) E3 ubiquitin ligases (Figure 2) [44,45]. This subsequent complex polyubiquitinates NRF2 and thus primes for proteasomal degradation, resulting in constitutive basal NRF2 turnover and low abundance [46,47]. As such, NRF2 has a short half-life of approximately 10–30 min [44,48,49]. The Yamamoto group established that this half-life could be extended in macrophages upon treatment with the electrophile diethylmaleate (DEM), inducing NRF2 target gene expression within 2 h, but through a post-transcriptional modulatory approach as NRF2 mRNA did not increase [48]. The KEAP1 protein is thiol-rich with many cysteines that can be targeted by electrophilic molecules such as DEM to perturb the NRF2-KEAP1 interaction [46,47]. Extensive research by the Dinkova-Kostova group has characterised unique KEAP1 cysteine groups and the functional consequences on NRF2 binding where Cys151 has been identified as one of the major stress-responsive residues [50]; more specific details of the NRF2-KEAP1 molecular interactions have been reviewed elsewhere [34,46]. Briefly, modulation of Cys151 and other residues by either exogenous stimuli or endogenous signalling molecules such as ROS does not perturb KEAP1 binding to CUL3 nor dissociate bound NRF2 from the KEAP1 dimer [34]; NRF2-activating molecules are capable of dissociating KEAP1-DLG interactions, but the “latch” ETGE mechanism remains associated despite stimulus intensity [35]. In fact, NRF2 under these circumstances occupies KEAP1 binding sites to allow out-titration by de novo NRF2 molecules, which are then free to undergo nuclear translocation and downstream transcriptional regulatory activity (Figure 2) [34]. The tight regulation and rapid nature of both NRF2 turnover and KEAP1 modification for NRF2 activation allow for highly sensitive and dynamic responses to intracellular environmental changes. NRF2 therefore serves as an important stress-related transcription factor with wide-ranging functions.

## 3. Cellular Functions of NRF2

### 3.1. Antioxidant Defence: The Canonical NRF2 Function

The classical function of NRF2, which has been understood since shortly after its discovery, characterises this protein as a master regulator of dynamic antioxidant responses [51]. An early study from the Yamamoto group reported how the knockout of NRF2 in mice induced high sensitivity to the hepatotoxic agent acetaminophen with concurrent loss of critical drug detoxification and antioxidant gene expression [52]. Furthermore, NRF2-null mice have consistently shown similar phenotypes of increased susceptibility to exogenous toxicity and oxidative disease-emulating conditions across multiple tissues including liver, lung, brain and stomach [53].

Our understanding of ROS signalling has expanded greatly beyond the simple hypothesis that these molecules solely activate oxidative stress pathways. It is now well understood that ROS, in physiological levels, also behave as signals to active a plethora of homeostatic mechanisms across almost all tissue types and, notably, in the maintenance of healthy cardiovascular functions [54]. For example, nitric oxide (NO) is one of the primary regulators of vascular tone which, through activation of a series of G protein-coupled receptor-mediated steps, causes the relaxation of endothelial smooth muscle cells [55] and thus facilitates blood pressure regulation [56,57]. Furthermore, superoxides, hydrogen peroxide and various other ROS molecules are central to key features of cardiovascular development and maintenance such as stem cell differentiation [55,58], angiogenesis [55], bioenergetics [57,59] and excitation–contraction coupling [59]. Cardiac tissue itself has a very high metabolic demand in comparison to many other tissues and organs given its critical function of pumping blood throughout the body [60]. For this reason, cardiomyocytes have a higher abundance of mitochondria, the organelle through which >95% of cardiac ATP production is facilitated [61,62], when compared to all other cell types at approximately 40% of the cell volume [62,63]. Thus, cardiomyocytes are well-equipped to support the vast aerobic metabolic demands of the myocardium [54]. However, the propensity for high ROS generation is also intrinsically linked to the mitochondrial density as ROS are by-products of normal mitochondrial functions [64] and, as such, these cells and tissues are particularly vulnerable to oxidative stress [54]. Despite this, cardiovascular cells can typically exhibit a lesser antioxidant potential compared to other organs. With respect to hepatic tissue, the enzymatic activity of numerous antioxidant defence proteins such as superoxide dismutases, glutathione peroxidases and reductases is significantly reduced in the heart [54,65]. Moreover, catalase activity was 50 times lower in heart than liver tissue samples [65]. Indeed, many common cardiovascular disease states have significant links to oxidative stress including heart failure [66,67], atherosclerosis [68], atrial fibrillation [69,70], myocardial fibrosis [71] and cardiac hypertrophy [67,72,73]. It is unsurprising, therefore, that the intrinsic links between oxidative stress and cardiovascular diseases prompted the interest in potential mitigation by NRF2.

Since the initial discovery and functional characterisation of NRF2 as a master regulator of antioxidant responses, it has become abundantly clear that its roles are multifaceted and extend far beyond this role. For reasons of brevity, this review will not provide a detailed overview of all understood mechanisms of NRF2 function (reviewed elsewhere [31,74,75]), but will outline key characteristics relevant to cardioprotection.

### 3.2. Iron Homeostasis

The regulation of iron is central to normal CV function, from the formation of haem groups for efficient oxygen transport and delivery via haemoglobin, maintenance of mitochondrial function, lipid metabolism, myocardial metabolism and erythropoiesis [76,77]. Furthermore, a lack of available iron is known to significantly increase the risk of developing cardiovascular impairments such as heart failure, coronary artery disease and pulmonary hypertension; approximately 50–70% of patients presenting with these diseases could be diagnosed with a clinical iron deficiency [76]. NRF2 is known to impact all aspects of intracellular iron homeostasis, as reviewed in detail here [77], from haem biosynthesis and catabolism, modulation of labile iron and control of ferroptotic cell death pathways—an iron-dependent mechanism [77]. Briefly, NRF2 drives the expression of a collection of genes that are iron-related. In a pioneering study by Campbell and colleagues using ChIP-Seq with subsequent pharmacological activation and genetic inhibition of NRF2, they identified and validated numerous direct NRF2 transcriptional targets in two lung-derived cell lines (A549 and BEAS-2B) and the liver-derived HepG2 cell line [78]. In all three cell lines, NRF2-silencing significantly reduced mRNA expression of ATP binding cassette subfamily B member 6 (ABCB6) by ~68–72%. Similarly, NRF2-knockdown reduced ferrochelatase (FECH) and heme responsive gene 1 (HRG1) mRNA levels across all three cell types, but most notably in HepG2 cells, suggesting a potential cell line or tissue-specificity to this NRF2-depenent regulation [77,78]. Moreover, ABCB6 and HRG1 protein expression was significantly increased upon 5 h sulforafane (SFN) treatment, a pharmacological NRF2 activator, and via KEAP1-silencing in erythroid K562 cells, further substantiating these results [78]. Both ABCB6 and FECH have roles within heme biosynthesis, whereas HRG1 is a protein transporter that shuttles labile heme from haemoglobin breakdown into the cytosol [77], highlighting the dual functions and thus tight control of this homeostasis afforded by NRF2.

As described earlier, NRF2 can also modulate the free iron content of cells. This intrinsic link may be partially understood due to the interplay between labile iron and its transitions between ferrous (Fe^2+)^ and ferritic (Fe^3+^) states. These cyclic redox reactions can lead to the generation of the powerful ROS molecules hydroxyl radicals via the addition of an electron to hydrogen peroxide during the Fenton reaction [79,80]. NRF2 is known to transcriptionally regulate genes that encode proteins for free iron storage and efflux [81] and thus can control iron and iron-derivate homeostasis, which is critical within the cardiovascular system.

### 3.3. Inflammation

NRF2 is also a powerful anti-inflammatory protein via several independent mechanisms. One of the canonical transcriptional targets of NRF2, heme oxygenase-1 (HO-1), catalyses the degradation of heme when associated with cytochrome P450 [82]. HO-1 mRNA levels negatively correlate with the expression of pro-inflammatory cytokines such as interleukin (IL)-6, vascular endothelial growth factor A (VEGFA) and activating transcription factor 3 (ATF3) [83]. Furthermore, in atherosclerosis, a typical cardiovascular disease associated with inflammation whereby lipid plaque formation within blood vasculature causes endothelial cell damage, dysfunction and subsequently the recruitment of inflammatory cells and release of cytokines, many experimental studies have acknowledged the fundamental anti-inflammatory roles of NRF2. Vascular cell adhesion molecule-1 (VCAM-1) is paramount to the initial development of atherosclerosis, and its activity has been suggested as an early indicator of at-risk vasculature for plaque formation; 82% of atherosclerotic lesions express VCAM-1 [84]. Pharmacological stimulation of HO-1 reduces VCAM-1 expression [85,86]. Moreover, overexpression of NRF2 via an adenoviral delivery system in human aortic endothelial cells significantly dampened TNF-α-induced VCAM-1 expression in a dose-dependent manner while simultaneously impairing the TNF-α-induced monocyte to endothelial adhesion [86,87].

Cardiac fibrosis can be characterised as a myocardial protection mechanism where components of the extracellular matrix (ECM) such as collagen are deposited in response to tissue damage from a variety of toxic insults [88,89]. Due to the non-regenerative nature of mature cardiomyocytes, excessive tissue damage can lead to fibrotic scarring and impaired cardiac functions through the reduced elasticity and contractility of the muscle walls and, thus, diminished ejection fraction and other markers of healthy cardiac status [89]. Ultimately, these fibrotic events often precede cardiomyopathies and heart failure. Fibrosis can certainly be attributed as an inflammatory mechanism as, upon cardiomyocyte damage and death, inflammatory signals are initiated that can activate fibroblasts [90] and recruit macrophages alongside mast cells [91], further exacerbating the initial insult [89]. A natural compound, chrysophanol, was found to upregulate NRF2 and downstream canonical target (HO-1, GCLC, GCLM) protein expression in the H9C2 rat cardiomyocyte cell line [92]. This was also associated with a dampened LPS-induced ROS and inflammation response as detected by fluorescence microscopy, Western blotting and RT-qPCR analysis [92], which was further corroborated during in vivo experiments. Using a high-fat diet (HFD) model of cardiac injury and NRF2^−/−^ C57BL/6J mice, this group suggested that a HFD significantly increased collagen accumulation (approximately 60% higher than control hearts), which could be mitigated through chrysophanol therapy [92]. However, in NRF2-knockout hearts, this reduction in collagen level was abolished, suggesting a dependence upon intact NRF2 signalling to drive this chrysophanol-induced pathway. The transforming growth factor-β1 (TGF-β1) has a well-established role within fibrotic advancement through promoting collagen expression [92,93]. HFD significantly increased TGF-β1 expression at both the transcriptional and protein level as well as downstream pro-collagen expression activators. Moreover, HFD significantly increased the expression of multiple pro-inflammatory cytokines (TNF-α and IL-1β, -18 and -6). Again, in NRF2-intact mice only, chrysophanol significantly suppressed the TGF-β1 and inflammatory mechanisms, resulting in less fibrotic heart tissue when compared to the untreated HFD mice [92]. To further support the direct causal links between murine NRF2 and their antifibrotic hypothesis, however, this study could have utilised the reportedly NRF2-specific pharmacological inhibitor ML385 [94] or siRNA alongside chrysophanol with the expectation that NRF2 inhibition would suppress the chrysophanol-induced fibrotic protection. This study has, notably, been supported by another group since its publication who proposed that NRF2 is a critical gatekeeper for TGF-β1-mediated Notch signalling (Notch4 contains an ARE and so is plausibly a putative NRF2 transcriptional target) and subsequently the development of fibrosis [95].

In a rat model of type II diabetes mellitus (T2DM), a disease where >70% of patients die from cardiovascular-related complications or disorders [96], cardiac NRF2 signalling was shown to decrease upon T2DM progression (reduced mRNA and protein levels alongside decreased target gene expression) with a concurrent gain of cardiac nuclear factor kappa-light-chain-enhancer of activated B cells (NF-κB) expression and function [97]; NF-κB is a transcription factor that modulates an inflammatory response upon its induction by powerful pro-inflammatory signals such as tumour necrosis factor-alpha (TNFα). The resulting pathology suggested that T2DM rats showed mitochondrial dysfunction hallmarks and highlights the potential crosstalk between NF-κB and NRF2 [98] to co-ordinate inflammatory responses in the diabetic heart [97].

It must be acknowledged that, as with most scenarios involving this transcription factor, NRF2 is not only an anti-inflammatory protein but, under differing physiological circumstances, can promote inflammation and immune responses. These mechanisms are more often triggered in early disease development such as in early myocardial ischemia-reperfusion, where macrophage polarisation and recruitment exacerbate the risk of injury and cardiomyocyte apoptosis and were suggested to be NRF2-dependent [99]. However, it can often be difficult to disseminate whether upregulated NRF2 regulation is causal or in response to cytotoxic stimuli, specifically due to its highly transient activity.

### 3.4. Mitochondrial Biogenesis

As is well-understood within the field, mitochondria are imperative for energy production and a wide array of cellular functions. The intricate structure of the mitochondrion, with an internal matrix surrounded by inner and outer membranes, permits the generation of electrochemical gradients to support the translocation of ions and small molecules to drive ATP synthesis in combination with metabolic pathways, which are reviewed in extensive detail elsewhere [100,101,102]. Mitochondrial biogenesis and the maintenance of structural integrity are crucial for the constitutive ATP output to meet cellular metabolic demands which, as discussed earlier, are highly required within the cardiovascular system.

Interestingly, the first connection between NRF2 and mitochondrial biosynthesis was suggested in 2008 by Piantadosi and colleagues during their research into the progression of cardiomyopathy [103]. In mice, they found that Akt activation inhibited GSK3-β and allowed NRF2 nuclear translocation to enhance nuclear respiratory factor-1 (NRF-1) expression [103]. NRF-1, not to be confused with the NRF2 family member, drives transcriptional expression of most genes, which encode the mitochondrial respiratory complexes that are required not only for biogenesis but also intact bioenergetics [104,105]. The transcriptional cofactors peroxisome proliferator-activated receptor γ coactivators (PGC)1α and 1β, which enhance the binding of NRF-1 to transcriptional targets for mitochondrial biosynthesis [106], are likely under the influence of NRF2 control. In fact, co-activation in the form of a positive feedback loop has been suggested as a point of crosstalk between NRF2 and PGC1α pathways with a consequential multifactorial activation of mitochondrial biogenesis and maintenance such as activating mitochondrial transcription factor A (TFAM), which contributes to mitochondrial DNA stability [107,108]. In this context, NRF2 can increase PGC1α activity and reciprocally undergo further activation by PGC1α. No physical protein–protein interaction has yet been described between these molecules [109,110], despite suggestions of co-localisation upon proteasomal inhibition [111]; however, this is not sufficient evidence to support an interactional hypothesis. Beyond this, NRF2 also regulates mitophagy and biogenesis through its influence on multiple mechanisms such as the p62-KEAP1 pathway in mitophagy [112]. Interestingly, NRF2 can promote mitochondrial biogenesis through the simultaneous suppression of mitochondrial fission and stimulation of fusion, as has been reviewed comprehensively elsewhere [113].

Taken together, these data support the role of NRF2 in maintaining cellular and metabolic homeostasis through balancing mitochondrial genesis, turnover, bioenergetics and dynamics [114].

## 4. NRF2 during the Progression of Cardiovascular Diseases

It is very common for NRF2 dysfunction to be measured in CVD given the aforementioned contribution of this transcription factor to cardiovascular homeostasis. As such, many studies have utilised genetic manipulation of the NRF2 signalling pathway to understand how blocking these mechanisms impacts the genesis and development of disease [115]. Homodeletion of *NFE2L2* in C57BL6J mice has been shown to considerably disrupt many cardiovascular outputs. Firstly, the hearts of NRF2^−/−^ mice displayed structural perturbations relative to WT animals, whereby a significantly increased heart weight and left ventricular mass relative to body weight were measured, along with indications of cardiac hypertrophy, which was non-inflammatory related [116]. Moreover, Erkens et al. measured significant changes within a number of cardiac functional readouts that suggest that NRF2 may be a key regulator of diastolic (but, intriguingly, not systolic) functions. They measured increases in deceleration time, mitral valve ejection time and myocardial performance indices with an accompanying significant reduction in protein expression of the sarcoplasmic/endoplasmic reticulum Ca^2+^ ATPase 2α (SERCA2α), indicating aberrant Ca^2+^ reuptake as a cause for the impaired myocardial relaxation upon NRF2^−/−^ [116]. Reduced SERCA2α expression and function have also been implicated in the development of heart failure [117]. The direct relationship between human NRF2 and SERCA2α has also been established by Val-Blasco and colleagues, whereby knockdown of NRF2 by short hairpin RNA significantly reduced SERCA2α protein expression in the human cardiomyocyte cell line AC16 [118]. These results were further supported by data from a constitutively active NRF2 mutant (NRF2^ΔETGE^; impairment of KEAP1-mediated proteasomal NRF2 degradation) having the opposing effect with significantly increased SERCA2α expression [118]. Taken together, there is convincing evidence that NRF2 is a key regulator of diastolic functions, which may be largely attributed to Ca^2+^ signalling.

### 4.1. Hypertension

In hypertension, perturbations in the relationship between NRF2 and oxidative stress has a key contribution to the increasing blood pressure. When ROS production exceeds the antioxidant capabilities of NRF2 activity, vasoconstriction can be triggered, perhaps via impaired nitric oxide (NO) signalling [119]. In a spontaneously hypertensive rat (SHR) model, NRF2 and its target gene (superoxide dismutase; SOD1, catalase, NQO1 and HO-1) expression were significantly downregulated but, when restored upon induction by L-sulforaphane, a significant reversal of the mesenteric atrial vascular dysfunction was observed [120].

### 4.2. Heart Failure

Heart failure is one of the largest global causes of morbidity and mortality with an estimated incidence of 1–2% in various Western regions and as high as 6.7% in Asian nations such as Malaysia [121]. Heart failure is a clinical nomenclature that covers various underlying pathologies but largely encompasses a condition whereby the heart becomes incapable of meeting the metabolic demands of the body, which can be measured by reducing cardiac output, tachycardia, faster onset of fatigue and impaired exercise tolerance [8]. Much focus is placed on understanding the biochemical, molecular and cellular mechanisms that underpin heart failure in order to better characterise the condition that would, in turn, facilitate the development of more efficacious pharmacological interventions of which current availability is limited [8,122]. A recent study into NRF2-KEAP1 signalling in dilated and ischaemic cardiomyopathies (DCM and ICM, respectively) revealed a significantly increased *NFE2L2*/*KEAP1* transcript ratio in both DCM and ICM patient samples with respect to healthy patients, though this may be driven more strongly by greater reductions in *KEAP1* expression than increasing *NFE2L2* [8]. Despite this marginal upregulation of *NFE2L2* mRNA expression, classical ARE-harbouring NRF2-target gene expression was predominantly reduced in the diseased samples [8]. Out of twelve target genes assessed, ten (83%) were significantly downregulated in at least one of the cardiomyopathic lineages with seven (58%) downregulated in both: NQO1, SOD1, glutathione peroxidase-3 (*GPX3*), glutathione peroxidase-4 (*GPX4*), glutathione reductase (*GSR*), peroxiredoxin 1 (*PRDX1*) and thioredoxin reductase 1 (*TXNRD1*) [8]. Notably, none within the selected panel of genes showed transcriptional upregulation in heart failure compared to healthy samples. Together, these results suggest that, despite the measurable increases in *NFE2L2* transcript expression and the increasing *NFE2L2*/*KEAP1* mRNA ratio, this was not translating into increased ARE target gene transcriptional upregulation, therefore NRF2 signalling was dysfunctional in DCM and ICM. Importantly, the results published by Lu et al. did not show how protein expression changed upon translation from the dysregulated mRNA. The consistent nature of the antioxidant and detoxification genetic target downregulation may imply changes at the protein level, however. Research into cardiovascular pathophysiological mechanisms, such as in this heart failure context, is unfortunately limited by the difficulty of obtaining human cardiac samples from patients and donors given the non-regeneration of the tissue, invasiveness of the surgery (will not be performed just for healthy control samples) and thus necessity for explanted hearts, which can be challenging to obtain [123]. This further highlights the requirement for translatable in vivo disease models.

Sprague Dawley rats with ligated left anterior descending coronary artery-induced chronic heart failure (CHF) displayed a significant increase in NRF2 mRNA with a significant decrease in cardiac NRF2 protein expression [124], further supporting the argument for the dysregulation of NRF2 transcription and translation in CVD. Moreover, a classical pharmacological inducer of NRF2, CDDO-Me, which modifies the Cys151 of KEAP1 resulting in NRF2 stabilisation [125], significantly improved the CHF stroke volume and cardiac output with the simultaneous restoration of NRF2 protein expression and reduced cardiac oxidative stress measured by confocal microscopy [124]. Furthermore, CDDO-Me mitigated the cardiac inflammation induced by CHF as measured by NF-κB activity [124]. This, again, highlights the multifactorial ability of NRF2 to provide cardioprotection.

### 4.3. Atherosclerosis

Paradoxically, NRF2 has also been shown to promote or exacerbate pathophysiological cardiovascular outcomes under different scenarios and appears to have a dual function in disorders such as atherosclerosis, obesity and T2DM, as reviewed in more detail here [86]. Atherosclerosis is often mitigated by NRF2 during early phases of its development [89]. Selective knockout of NRF2 targets, such as GPX1 [126] and HO-1 [11,127,128] in vitro and in vivo, routinely promote atherosclerotic progression, amplified immune responses and plaque formation [86]. Contrastingly, the deletion of murine NRF2 impaired diet-induced atherogenesis with reduced IL-1 production and inflammation [129]. To support this, NRF2-knockout in apoliporotein E-deficient (ApoE^−/−^) C57BL/6J mice showed a significantly reduced aortic plaque size relative to the NRF2^+/+^ ApoE^−/−^ controls [130]. Flow cytometry techniques found that macrophages isolated from the NRF2-deficient group exhibited significantly impaired low density lipoprotein (LDL) reabsorption, a key function of macrophages in the cardiovascular system [130]. This reduced capacity may be due to the expression of CD36, an LDL-recognition receptor that positively correlates with plaque development and appears to be driven by NRF2 under stressed conditions [130].

### 4.4. NRF2 Contribution to Cardiac Dysfunction

These dual pathogenic functions of NRF2 are further exemplified following clinical trials of the NRF2 activating CDDO-Me which, as discussed earlier, showed promise in the treatment of chronic kidney disease (CKD) during preclinical and phase II trials. However, the phase III BEACON clinical trial for CKD was terminated primarily due to the increased risk of heart failure requiring hospitalisation in the CDDO-Me group when compared to the placebo (8.8% and 5.0% of patients, respectively [131]). Following this termination, further analyses were conducted to observe the CDDO-Me mediated effects, which revealed that this NRF2 agonist reduced endothelin signalling in the renal medulla, which prominently modulates both sodium and water homeostasis whereby CDDO-Me increased its retention relative to the placebo group; this led to an increased blood pressure in the treatment group [132]. However, it was suggested that the differential fluid handling mediated through CDDO-Me therapy may be distinct from the heart failure findings as no correlation was measured between these outcomes [132]. Furthermore, there has been sufficient evidence to suggest that the prominence of increased fluid retention from CDDO-Me treatment was only measured in patients with stage 4 and 5 CKD, who are already much more susceptible to renal physiological changes due to pre-existing dysfunctional endothelin signalling [132,133]. Moreover, the systolic and diastolic blood pressures were increased in stage 4 but not stage 3 CKD, further implying the correlation between CKD severity and CDDO-Me mediated cardiovascular complications [132,133].

These findings substantiate the general hypothesis presented in this review that acute or transient NRF2 activation during early disease development may help preserve healthy homeostatic function whereas chronic activation, or activation at a more advanced disease stage, cannot reverse previous damage and may exacerbate the pre-existing complication. The tight regulation of NRF2 under physiological circumstances, predominantly via KEAP1-coordinated proteasomal degradation and thus short protein half-life, may be an intrinsic mechanism that has been optimised to extract the cytoprotection afforded by NRF2 without exposing tissues to prolonged NRF2 activity and thus increased risk of deleterious chronic activation.

## 5. NRF2 in Ageing-Associated Cardiovascular Diseases

Ageing phenotypes were acquired over time for numerous biological reasons which, were categorised into nine groups in 2013 as the “hallmarks of ageing”, including mitochondrial dysfunction, genomic instability, cellular senescence and loss of proteostasis [134]. Furthermore, one of the most highly influential concepts within the field of physiological ageing is the progressive damage caused by the accumulation of pro-oxidative molecules, leading to impaired tissue function [135]. Over time, these detrimental effects can result in age-related diseases, many of which have clear known oxidative stress components to their initiation and development including renal diseases, diabetes and, for the context of this review, CVD [135]. As a master regulator of dynamic antioxidant responses, NRF2 has been suggested as a potential mitigator of the oxidative stress-linked ageing process, as summarised in detail by Schmidlin and colleagues [136]. The interplay between NRF2, ageing and the consequences for CVD has also been thoroughly reviewed elsewhere [137], therefore this review will summarise some of the key findings.

### 5.1. In Vivo Models of Ageing

There are several animal models well-suited to investigation into the prevention of age-related diseases and the extension of longevity. The naked mole-rat (*Heterocephalus glaber*) is often studied given its greatly increased captivity life span relative to other rodents and mammals of a similar body size, living up to 31 years [138]. To elucidate the potential mechanisms driving the increased longevity of this species, Lewis et al. hypothesised that increased NRF2 signalling in the naked mole-rat was providing greater resistance to cellular stressors and thus mitigation from age-related diseases, leading to an increased lifespan [138]. Intriguingly, increasing the maximum lifespan potential was positively correlated with stronger liver NRF2-ARE binding activity when measured across nine rodent species. Moreover, this group reported a significant upregulation of NRF2 mRNA and protein with a concurrent loss of KEAP1 at the protein and transcriptional level [138] in naked mole-rats when compared to the shorter-lived rodent *Mus musculus*. The expression and activity of downstream NRF2 targets including GST and NQO1 were also increased significantly (2–5-fold increases in activity) in the naked mole-rat [138].

Comparative studies have been published in multiple model organisms including bats, birds and invertebrates [139] with consistent findings that NRF2 (or its orthologs) signalling is impaired in the ageing process. A recent study in *Drosophila* highlighted an interesting phenomenon that is common within NRF2 dynamics. Whilst mild CncC activation (NRF2 ortholog) significantly lengthened the lifespan with increased associated NRF2-related downstream cytoprotective gene expression, hyperactivation of the NRF2 pathway (e.g., by KEAP1 suppression) actually stunted the longevity as cellular resources were potentially reallocated to combat the persistent stress [140]. Moreover, this group reported the concurrent increased expression of type 1 diabetes and aberrant mitochondrial bioenergetics markers [140]. A comprehensive review by Zhang and colleagues suggested an age-related perturbance of redox homeostasis, largely driven by increased oxidant production and a simultaneous impairment of antioxidant signalling with increased age [141]. Primate research published in 2011 measured significantly increased carotid artery oxidative stress and inflammation in old (22 years) versus young (10 years) Rhesus monkeys [142]. Moreover, a non-significant suppression of arterial NRF2 protein expression and nuclear binding activity was measured in the aged group, but the inducibility of NRF2 by H_2_O_2_ in vascular smooth muscle cells was significantly impaired with age [142]. It may be this loss of NRF2-driven adaptive and dynamic responses to cellular stresses including oxidative stress that leads to cellular senescence and tissue ageing [143], which may result in the subsequent increase in age-related disease phenotypes such as CVD (Figure 3).

### 5.2. NRF2 Signalling during Human Ageing

A study by Safdar et al. has also suggested impaired NRF2 activity from the skeletal muscles of sedentary, but not active, elderly humans, which was again attributed to a loss of dynamic adjustment to pro-oxidative stimuli as measured by the increased oxidant damage and depleted glutathione in the sedentary adults [144]. These results were corroborated more recently in a study of human bronchial epithelial cells from young (21–29 years) and older (60–69 years) subjects [145]. Here, this group reported an elevated basal transcriptional expression of the typical NRF2 target genes *GCLC*, *GCLM* and *NQO1* in cells from older donors, but with reduced pharmacological inducibility, which further supports the hypothesis that NRF2 dynamic responses are perturbed during the ageing process [145]. The increased baseline expression was suggested to be non-NRF2 driven, however, due to the downregulated basal and inducible NRF2 protein expression in older cells with an associated decline in activity as measured via an electrophile response element (EpRE)-luciferase reporter assay [145]. The results suggested by Safdar and colleagues would be interesting to translate into a cardiovascular capacity to assess whether vascular smooth muscles and myocardial tissue similarly display these changes to NRF2 signalling as measured in skeletal tissue. Taken together, there is substantial evidence to suggest that dynamic NRF2 responses that would ordinarily provide healthy cells with protective mechanisms to combat an array of cellular stressors become dysregulated during the ageing process (Figure 3), which has been measured almost ubiquitously across many experimental animal models and human samples.

### 5.3. Cardiovascular Diseases and Ageing

The prevalence of CVD is also highly correlated with increasing age. According to a 2009 report by the American Heart Association (AHA), the incidence rates of CVD between the ages of 40–59 were approximately ~38%, increasing to ~73% and ~79–86% in age groups 60–79 and ≥80, respectively [146]. Similar trends of increasing prevalence have also been reported for multiple cardiovascular co-morbidities including myocardial infarction, congestive heart failure and coronary heart disease [146]. Given the globally increasing ageing population and rising prevalence of many cardiovascular disease risk factors such as obesity and diabetes [147], CVD will remain a continuing and growing health issue of intense importance and thus requires the attention of scientific research. In a recent study of cardiovascular function decline in ageing mice, ejection fraction, stroke volume and cardiac output were all significantly reduced in older animals when compared to younger equivalents [148]. Concurrently, the binding of NRF2 to ARE sequences was reduced by approximately 50% in heart tissue from older mice and the NRF2 target gene transcriptional expression was significantly downregulated for several canonical targets including antioxidant and anti-electrophile enzymes [148]. Cardiac NRF2 activity was, however, completely restored in the older mice through SFN treatment with a simultaneous improvement in cardiac function. One explanation for this could be the prevention or reversal of age-related declining cardiac mitochondrial integrity and activity through the upregulation of NRF2 [148]. Clearly, targeting this pathway poses an efficacious avenue through which ageing within the cardiovascular system could be mitigated.

## 6. Drug-Induced Cardiotoxicity and the Role of NRF2

Changes within intracellular biology can lead to most forms of CVD and cardiac disorders, as has been discussed in this review. However, these phenomena can also be triggered from exogenous stimuli such as pharmaceutical agents, many of which have well-understood cardiotoxic side effect profiles. Notoriously, the anthracycline class of anticancer drugs, including doxorubicin (DOX), daunorubicin and epirubicin [149], have clear cumulative dose-dependent toxicity, which can present in a plethora of clinical manifestations and unfortunately tend to be irreversible and often life-threatening [150]. Moreover, cardiotoxic events may only appear acutely or years, even decades, beyond treatment cessation [151]. In the case of DOX, despite being discovered in the 1960s, it is still routinely utilised today for the treatment of multiple cancer types, commonly for haematological cancers or solid tumours such as of the breast and ovaries [152,153] and in up to 60% of childhood cancers [154].

### 6.1. Pathophysiology of Doxorubicin-Induced Cardiotoxicity and Antitumour Activity

The diagnosis of cardiotoxicity is often difficult, partly due to inconsistent definitions and thresholding of measurable outcomes such as left ventricular ejection fraction (LVEF) reductions [155]. In a comprehensive analysis published by Swain et al., data from three independent doxorubicin treatment trials validated the DOX cumulative dose relationship with cardiotoxicity whereby patients receiving a cumulative dose up to 600 mg/m^2^ had an 8.7% chance of a congestive heart failure diagnosis, whereas this prevalence was 1.6% and 3.3% for ≤300 mg/m^2^ and ≤450 mg/m^2^, respectively [156,157]. Even at previously perceived low cumulative doses of ≤250 mg/m^2^, the risk of adverse cardiac incidents was increased by 7.8–8.8% [158] and induced defects in endomyocardial biopsies [159]. As such, a maximal lifetime exposure has been recommended at 450–500 mg/m^2^ [160,161].

Despite its longstanding clinical utility and well-understood cardiotoxic profile, investigations into the biological alterations induced by DOX that lead to the associated damage are still ongoing and can be somewhat controversial [162]. The delineation of off-target effects is also intrinsically linked to the mechanism of action for the antineoplastic activities of DOX. Firstly, upon nuclear import, DOX intercalates within DNA, forming covalent bonds between the two double helix strands, which causes unwinding, tortional stress and the activation of DNA damage events [162]. Synergistically, DOX inhibits topoisomerase IIA (TOP2A) in the fast-dividing cancer cells and thus interferes with DNA replication and repair mechanisms. This hypothesis has been supported through in vivo evidence that TOP2A expression can dictate DOX efficacy in a murine lymphoma model whereby the knockdown of TOP2A conferred DOX-resistance [163]. However, there are multiple mechanisms of cell damage that extend beyond the DNA. DOX is known to induce oxidative stress, which was believed to be a primary cause of its antitumorigenic activity following measurable increases in ROS from animal and human studies [164]. However, more recently it has been proposed that ROS formation in itself does not make a large contribution to its on-target mechanism of action due to the relative resistance of many cancer cell types to oxidative stress, as will be discussed later in this review, and the requirement for often supra-clinical DOX doses to induce excessive ROS levels [161,165]. The DOX-induction of ROS may provide a secondary mechanism for its anticancer efficacy, largely due to the support of downstream intracellular pathway aberrations in response to this imbalanced redox environment. These can include iron dysregulation, chromatin damage, metabolic changes and the induction of regulated cell death pathways [161,165,166].

Unfortunately, the delivery of DOX does not selectively target cancerous tissues, but also exposes the cardiovascular system to many opportunities for these same mechanisms to cause toxicity to healthy cells and tissues. Complexed with this, there are several cardiac-specific vulnerabilities to DOX that often lead to the development of off-target cardiotoxicity. Cardiac tissue does not express TOP2A, but has a high abundance of TOP2B [167]; DOX targets both isoforms and therefore the inhibition of DNA repair, initiation of double strand breaks and stimulation of DNA damage mechanisms leading to apoptosis and other death pathways in non-cancerous cells [168]. DOX has also been understood to disrupt cardiac bioenergetics and may also induce antineoplastic properties in a similar fashion. A more in-depth review of DOX and mitochondrial contributions to cardiotoxicity can be found here [169]. Briefly, DOX accumulates strongly within mitochondria and typically disrupts electron transport chain elements and thus interferes with respiration and can increase ROS synthesis with downstream consequences [169,170], further referring to the aforementioned synergistic, but not primary role of oxidative stress in the pathogenesis of DOX-induced cardiotoxicity. Moreover, DOX disturbs the homeostatic balance between mitophagy and biogenesis that is required for the maintenance of efficient and numerous mitochondria, which is especially important with respect to the cardiovascular system. Using the human cardiomyocyte cell line AC16, it has been shown that DOX promoted mitophagy, inhibited biogenesis and ultimately reduced cell viability, which was attributed to the activation of the PTEN-induced kinase 1 (PINK1)/parkin pathway [171], a well-understood quality control pathway that facilitates the removal of damaged mitochondria [172]. Interestingly, NRF2 regulation of PINK1 is an emerging topic, insofar as PINK1 may even contain up to four ARE regions within its promotor sequence, which would presumably allow NRF2 binding and transcriptional regulation [173]. NRF2 pharmacological induction and genetic knockdown experiments further supported this hypothesis given that increased NRF2 expression correlated positively with PINK1 expression [173].

### 6.2. Doxorubicin Cardiotoxicity and NRF2 Pathway Interplay

What is clear from this review is the high degree of overlap between mechanisms of DOX-induced toxicity and NRF2 regulation (Figure 4). The manipulation of NRF2 pathways has, therefore, emerged as one of the key research topics for the alleviation of DOX-induced cardiotoxicity. As opposed to cancer cells, cardiomyocytes are highly sensitive to unstable oxidative environments, as discussed earlier. It is not surprising, therefore, that the upregulation of NRF2 expression, as shown by Zhang et al., in either mice or isolated cardiomyocytes, reduced the DOX-induced oxidative stress phenotypes (increased ROS, reduced SOD activity) while reversing the DOX-induced apoptotic pathway stimulation as measured through caspase, BAX and BCL-2 expression [174]. Ferroptosis, the iron-dependent non-apoptotic cell death pathway, can also be inhibited upon NRF2 activation in DOX-induced cardiac tissue. Hearts from DOX-treated rats and H9C2 cardiomyocytes showed decreased GPX4 expression, one of the key regulators and inhibitors of ferroptosis while concurrently reducing cell viability [175]. Similarly, DOX significantly reduced NRF2 expression both in vitro and in vivo, which was stabilised when co-treated with fisetin, despite fisetin monotherapy not inducing NRF2 expression above control levels itself, and so may act as an NRF2-stabiliser rather than a pharmacological inducer [175]. Fisetin could also restore GPX4 expression in a dose-dependent manner while attenuating DOX-induced myocardial dysfunction in the rat heart and significantly improved cell viability in response to DOX [175]. In support of these results, Wang et al. recently reported the contribution of NRF2 to the alleviation of DOX-induced cardiotoxicity via ferroptosis with a focus on protein arginine methyltransferase 4 (PRMT4), a transcription factor also involved in autophagy and redox balances [176]. This group observed that direct interactions between PRMT4 and NRF2 sterically hinder the nuclear translocation of NRF2 and therefore diminish GPX4 expression. Consequently, PRMT4 overexpression exacerbated doxorubicin-induced cardiotoxic phenotypes such as reduced LVEF and increased myocardial fibrosis [176]. Both the pharmacological activation of NRF2 (CDDO-Me) and ferroptosis-specific inhibition with ferrostatin-1 completely ameliorated the toxicity of DOX to these cardiovascular cells [176].

As has been discussed earlier, NRF2 is intrinsically linked with cellular ageing processes. One notable mechanism that largely contributes to tissue ageing is cellular senescence; a process by which cells undergo physiological changes with increasing age to ultimately reduce metabolic activity, functional capacity and induce cell cycle arrest [177]. This process can be triggered under homeostatic conditions in response to internal stimuli, but may also be induced through pathophysiological means; cardiomyocyte senescence is a hallmark of DOX-induced cardiotoxicity and premature cardiac senescence may be a key aspect of late-onset DOX-toxicity [178]. Recently, Lerida-Viso and colleagues demonstrated that the hearts of mice repeatedly exposed to DOX over a four week period with a 20 mg/kg cumulative dose, thus mimicking a clinical treatment regimen, significantly overexpressed senescence markers such as p16, p21 and p53 with the simultaneous impairment of cardiac function as measured via echocardiography [179]. These effects were reversed upon treatment with a senolytic agent [179].

Upregulation of NRF2 has been suggested as a mechanism for reduced cellular senescence in models of diseases such as osteoporosis [180], age-related macular degeneration [181], age-related cognitive decline [182] and renal fibrosis [183]. Recently, Liu et al. described how NRF2 is activated during DOX-induced senescence in two human cancer cell lines and in vivo by an accumulation of the oncogene iASPP (Inhibitor of Apoptosis Stimulating Protein of P53), which binds and inhibits KEAP1 [184,185]. They measured a significant increase in M2 macrophage polarisation upon DOX-induced cell cycle arrest, which was stimulated by upregulated iASPP [184]; immune cells are known to clear senescent cells [186]. This iASPP-induced macrophage polarisation was subsequently associated with increased DOX resistance as assessed by an increasing tumour volume size in apoptosis-resistant colon cancer xenograft mice models [184]. Further, this group proposes a novel NRF2 genetic target (macrophage colony-stimulating factor; M-CSF), a known inducer of macrophage polarisation, as a key contributor to the iASPP-NRF2 axis and thus to the chemotherapeutic resistance within tumours [184]. Theoretically, therefore, if NRF2 can inhibit DOX sensitivity within cancers by interfering with senescence pathways, it could be postulated that this pathway could be applied within the cardiovascular system to alleviate DOX-induced cardiac senescence and off-target cardiotoxicity.

Other regulated cell death pathways have also been implicated in DOX-induced cardiotoxicity such as pyroptosis, a mechanism distinguished by increased caspase activity, which results in plasma membrane pore formation following Gasdermin D/E (GSDMD/E) cleavage and the subsequent inflammatory response upon IL-1β and IL-18 release [166]. NRF2 overexpression has recently been shown to reduce pyroptotic markers (cleaved-caspase-1 and GSDMD) in microglia and thus improve cell viability in an LPS-stimulated pyroptotic model [187]. In a myocardial ischaemia-reperfusion (I/R) injury model, pyroptotic cell markers were significantly increased alongside a greater infarct size in the rat heart [188]. Treatment with the natural compound sweroside blunted these outcomes in a dose-dependent manner. Furthermore, sweroside promoted the nuclear accumulation of NRF2 proteins and inhibited hypoxia/reoxygenation-induced pyroptosis in the H9C2 cardiomyocyte cell line and confirmed the NRF2-specificity through siRNA transfection [188]. It must be noted that the factors that drive cells towards specific cell death pathways preferentially over others are unknown, yet experimentally isolating one pathway alone may be too simplistic and not representative of the true cellular physiology. Multiple cell death mechanisms may be activated simultaneously [189]. However, given the evidence that NRF2 has cytoprotective roles in most death pathways, global cellular NRF2 activation could be considered a potential therapeutic strategy for attenuating DOX-induced cardiotoxicity.

The direct effects of DOX on NRF2 expression and activity are often not primary objectives for research studies, but can be a point of contention within the literature. Conceivably, a drug that provokes such a ubiquitous disturbance to the cellular homeostasis, cytotoxicity and typical induction of oxidative stress would be expected to induce the stress-responsive NRF2 pathway. Despite this, many studies exhibit significant downregulation of NRF2 protein expression with the concurrent loss of target gene expression in response to DOX treatment. Myocardial NRF2 mRNA expression was significantly downregulated by >2.5-fold in rats receiving four weekly doses of 2 mg/kg DOX [190]. Similarly, murine ventricular tissue exhibited a 40% reduction in NRF2 protein expression following a 15 mg/kg cumulative dosage [191]. Others have reported no significant changes in NRF2 protein expression upon DOX treatment, again in the mouse heart [192]. In contrast, NRF2 expression measured through Western blotting and immunohistochemistry has been shown to significantly increase in C57BL/6J cardiac tissue following a single, 25 mg/kg DOX injection [193]. Additionally, primary neonatal rat ventricular myocytes exposed to 2 μM 24 h DOX exhibited an increased nuclear NRF2 protein accumulation [176]. Evidently, a common discrepancy that divides the direction of NRF2 expression in response to DOX is the acute or repeated nature and strength of treatment. Single, high doses appeared to induce NRF2 protein levels, but recurrent, lower dose therapy, which more accurately emulates patient dosing regimens, generally suppresses NRF2 expression and thus could be considered more clinically translatable.

Furthermore, the utility of non-clinically relevant doses of DOX is a widescale issue within this research community and may be indicative of a scientific failing in other areas whereby, in the pursuit of biological understanding, extreme conditions are induced, and therefore any findings acquired from these experiments may not be physiologically relevant, which must always be the ultimate aim for research. Notably, studies on doxorubicin plasma concentrations in breast cancer patients highlight how C_max_ levels can be measured between 300 nM [194] and 1 µM [195,196], therefore conceivably any dose ≤1 µM could be deemed clinically relevant. Some have reported higher peak DOX plasma concentrations, up to 6.9 µM, but this is only at the highest prescribed dose of 75 mg/m^2^ and very rapidly (within 10 min) returns to the sub-1 µM range [197]. Treating isolated cells for 24–48 h with 10 or even 100 µM DOX concentrations, as can often be found within the literature [195], could be exposing them to a clinically inappropriate dose with markedly different mechanisms of toxicity [198]. Unfortunately, it is therefore very difficult to translate in vitro results to more clinically relevant models.

## 7. Oncological Roles of NRF2

### 7.1. Somatic Mutations within NRF2 Regulation

A very important aspect of NRF2 signalling, especially in an anthracycline-induced cardiotoxicity framework, is the contrasting impact of NRF2 activity on cancer biology. NRF2 upregulation is a well-established mechanism through which many human cancers have increased aggression and malignancy through tumour proliferation and survival via therapeutic resistance [199] and immune evasion; such cancers are often characterised as “NRF2-addicted” [200,201]. The modulation of NRF2 activity is predominated by the adjustment of NRF2 protein stability and degradation as opposed to transcriptional expression changes of the *NFE2L2* gene, therefore somatic mutations within genes that encode NRF2 regulatory proteins such as KEAP1 (*KEAP1*), β-TrCP (*BTRC*) and CUL3 (*CUL3*) are common within NRF2-addicted cancers. Similarly, mutations within the NFE2L2 gene can perturb the interactions of NRF2 with its key regulators, thus providing a constitutively active NRF2 protein. A comprehensive genomic analysis of squamous cell lung cancers revealed that 34% of these cancers harboured somatic mutations in *NFE2L2* (19%), *KEAP1* (12%) or *CUL3* (7%), resulting in the downregulation of NRF2 repressors and activation of NRF2 [202]. While the distribution of *KEAP1* mutations are rather ubiquitous across the amino acid structure of the protein, the majority of *NFE2L2* mutations reside within the DLG and ETGE KEAP1-binding sequences of the NRF2 Neh2 domain [203]. NRF2 dysregulation via somatic mutations have also been implicated in ovarian, colon, breast, prostate and gastric cancers [204,205,206,207,208] and, as such, NRF2 inhibitors are being explored as therapeutic options where these signalling pathways have been constitutively activated [209].

### 7.2. NRF2 Signalling in Cancer

An important paradigm, which was introduced earlier in this review, whereby NRF2 activity undertakes different roles depending upon the stage of disease development, is classically true in cancer biology. This double-edged sword rationale, as described by Wu et al. [210], suggests that NRF2 can be both anti- and pro-tumorigenic, becoming overactive as the cancer progresses and matures [199,211]. NRF2-deficiency in mice increases the number of tumours in multiple organs [210]. In a mammary carcinoma model, an NRF2^−/−^ genotype markedly increased the tumour volume and weight while significantly reducing the tumour-free survival with respect to the WT group [212]. Additionally, pharmacological NRF2 induction by CDDO-Me significantly reduced the rate of tumour development in breast cancer-associated gene 1 (*BRCA1*)-deficient mice through the knockdown of ErbB2 protein expression [213], a known oncogene [214], with direct modification of ErbB2 cysteine residues [213]. This study did not measure NRF2 changes in response to CDDO-Me but, given the robust data presented within this review of CDDO-Me-mediated NRF2 stabilisation, it could be postulated that the increased NRF2 target gene expression may also have attributed to antitumorigenic properties. Multiple other compounds that upregulate NRF2 also blunt the oncogene transcription and exert anticancer activity [210].

### 7.3. Pharmacological Manipulation of NRF2 Signalling in Cancer Therapy

Tumorigenic development and subsequent survival places progressively higher metabolic demand upon the cell with increased mitochondrial activity, ATP supply and ultimately ROS production [215]. This excess ROS must be mitigated to prevent apoptotic cell death and enable the continued proliferation of the cancerous cell [215]. Hence, the promotion of NRF2 signalling may be favoured in these cells for adaptation to this increasing oxidative burden and provide an environment permissive to cell division and survival [215]. The dual oncogenic and tumour-suppressant nature roles of NRF2 appear to be highly dependent on multiple factors including the tissue type and, as discussed, the tumour developmental stage. Unfortunately, the discovery that many forms of cancer can harbour somatic mutations resulting in overexpression of NRF2 and subsequent tumour resistance makes the recommendation of pharmacological NRF2 stimulators to alleviate off-target effects of cardiotoxic drugs, notably the anthracycline class of chemotherapeutics, difficult. Therefore, it may be pertinent to stratify patients for whether this strategy poses more benefits that outweigh potential harms whereby if the patient presents with an NRF2-overexpressed tumour and clinicians are to prescribe, e.g., doxorubicin, an NRF2 activator may be a useful strategy to limit the cardiotoxic risk as no further benefit will be afforded to the neoplasm. Contrastingly, if tumoral NRF2 activity is comparable to healthy tissues with low physiological levels, NRF2 activation may not be recommended due to the increased risk of supporting therapeutic resistance within the cancer. However, if a cardiac-specific delivery strategy for an NRF2-inducer could be devised that exposes the myocardium to levels sufficient to stabilise NRF2 and yet avoid this outcome in cancerous tissues, this could still provide utility. Moreover, one of the key concepts of chemotherapeutic development is looking for cancer-specific signatures, which would facilitate the cancer-specific delivery of the antineoplastic drug, although this is highly complex and may still affect the cardiovascular system due to the high vasculature of most tumours and often intravenous delivery of chemotherapeutic medicines.

## 8. Conclusions

The maintenance of a healthy and functioning cardiovascular system requires constant dynamic adaptations to a variety of stimuli which, when dysregulated, can interfere with homeostatic physiology and lead to the development of diseases. Susceptibility to detrimental cardiovascular events can also be influenced by ageing processes, comorbidities including T2DM and drug-induced toxicity such as with anthracycline chemotherapy. The transcription factor NRF2 is often considered as a master regulator of stress responses and its activation has been explored as a therapeutic avenue by which cardioprotection can be achieved. Abrogation of NRF2 signalling often precedes or exacerbates the development of cardiovascular diseases. An increasing body of literature is supporting the hypothesis that NRF2 upregulation, particularly during the mitigation of early cellular damaging events, decreases the vulnerability of cells to impairment and thus retains carlidiovascular integrity. Over the coming years, increased research into the NRF2 gene expression network could highlight novel strategies for therapeutic interventions in cardiovascular disorders.

## Figures and Tables

**Figure 1 ijms-24-06723-f001:**
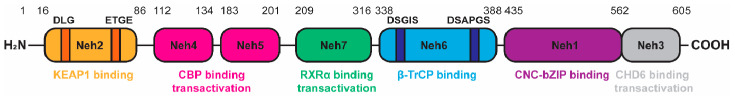
NRF2 protein structure with seven NRF2-ECH homology (Neh) domains, each permitting direct protein–protein interactions with key NRF2 regulators. Numbers correlate with the NRF2 amino acid sequence. Four Neh domains (Neh4, Neh5, Neh7 and Neh3) are transactivation domains. Neh2 harbours two binding motifs for KEAP1 interactions with markedly stronger affinity at ETGE than DLG. Neh6 similarly contains two degron sequences that facilitate β-TrCP binding; DSGIS and DSAPGS. Both KEAP1 and β-TrCP are negative regulators of NRF2 and facilitate its proteasomal degradation. Neh1 is a conserved domain with other bZIP proteins and permits the dimerisation of NRF2 with sMAF proteins for downstream binding to genetic targets.

**Figure 2 ijms-24-06723-f002:**
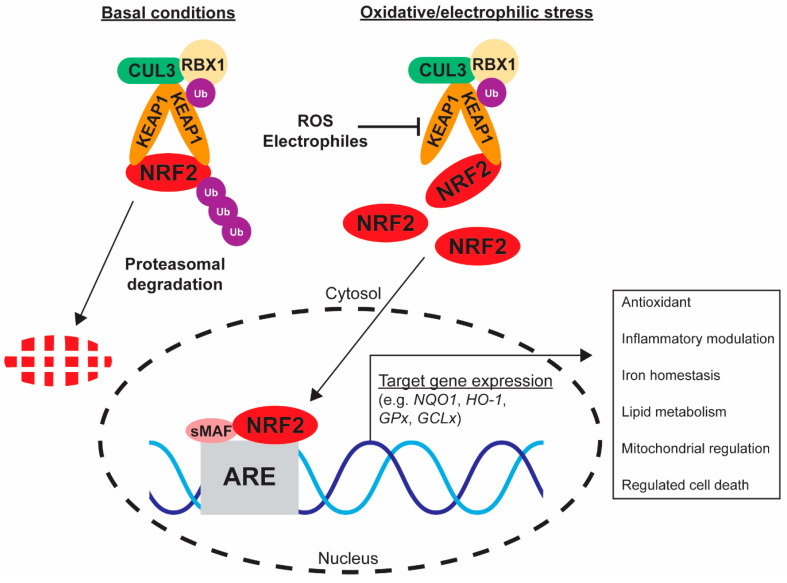
KEAP1-mediated regulation of NRF2 under basal and stressed intracellular environments. Basally, NRF2 is bound by a KEAP1 homodimer and retained within the cytosol. KEAP1 forms an E3 ubiquitin ligase complex via associations with CUL3-RBX1 and ubiquitinates NRF2, priming for subsequent proteasomal degradation. Increased ROS or electrophilic modification of KEAP1 cysteine residues perturbs the NRF2–KEAP1 interaction. De novo NRF2 is able to translocate to the nucleus, heterodimerise with sMAF proteins and bind to antioxidant response element (ARE) genetic sequences to drive target gene expression.

**Figure 3 ijms-24-06723-f003:**
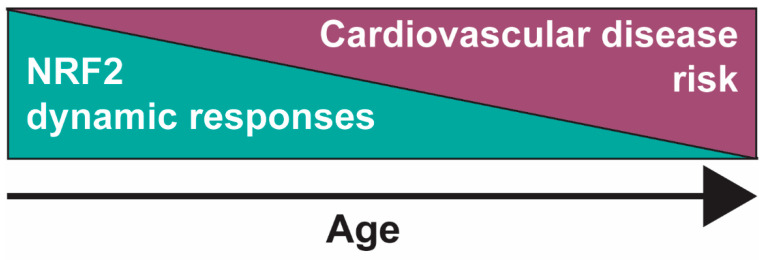
The propensity for NRF2-mediated coordination of dynamic responses to intracellular stressors such as increased ROS, xenobiotics or cardiotoxic drugs is decreased during physiological ageing. Cells of the cardiovascular system are therefore more susceptible to accumulative damage and tissue dysfunction, progressing towards disease phenotypes.

**Figure 4 ijms-24-06723-f004:**
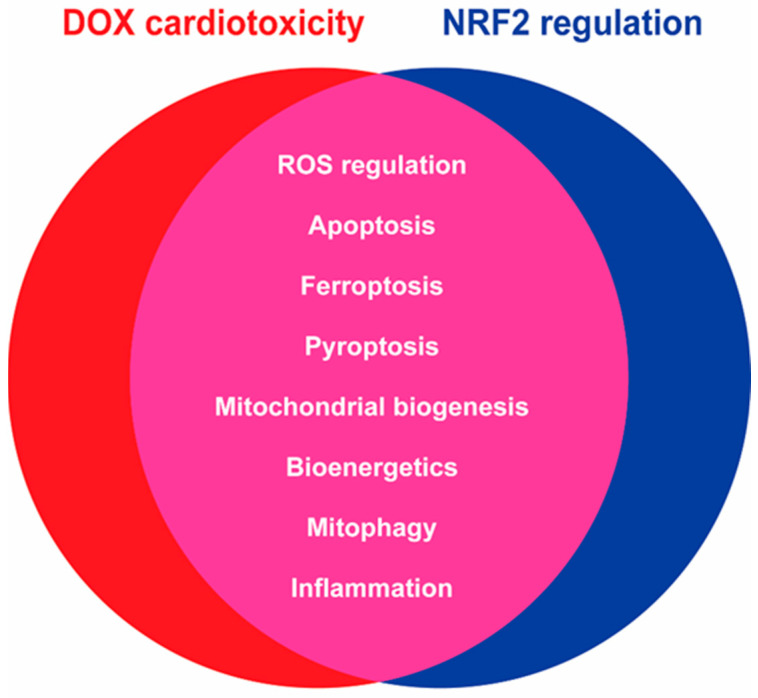
Interplay between mechanisms through which doxorubicin (DOX) induces deleterious off-target cardiotoxic events and pathways regulated under NRF2 transcriptional control.

## Data Availability

No new data were created or analyzed in this study. Data sharing is not applicable to this article.

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
