# Peer review of "Mitigation of Cardiovascular Disease and Toxicity through NRF2 Signalling"

_ijms, 2023, doi:10.3390/ijms24076723_

Round 1

Reviewer 1 Report

The authors reviewed the current knowledge of NRF2 on cardiovascular under normal and pathophysiological circumstances. I have a few major suggestions that the authors might consider.

1.     There are many better and well-constructed reviews in this area. Please see:

-NRF2 in Cardiovascular Diseases: a Ray of Hope!

-Nrf2 for protection against oxidant generation and mitochondrial damage in cardiac injury

-Nrf2 in aging – Focus on the cardiovascular system

-https://doi.org/10.1155/2017/9237263

-Nrf2 for cardiac protection: pharmacological options against oxidative stress, and so on

2.     The authors reviewed general issues on Nrf2. The review has to be rewritten in a more constructed and deep manner related to cardiovascular diseases.

3.     Why authors included the Cancer section?

4.     It is important to update this review with recently published articles. Please do a systematic literature review search to not miss related articles.

5.     Please include your search strategy.

6.     Please check the manuscript for typos or grammatical errors. Some sentences are not clear. 

Reviewer 2 Report

The review is well written, well organized, comprehensive and easy to read and understand.

Author Response

Response to Reviewer 2 Comments

We acknowledge that Reviewer 2 did not present further comments but we would like to thanks them for their time reading and assessing the manuscript.

Reviewer 3 Report

In this review, the authors provided an overview of the role of NRF2 The Role of NRF2 in Cardiovascular Function, Diseases, and anthracyline-induced cardiotoxicity. Overall, I found the article to be well-written and informative and the authors cited many seminal publications relevant to the topic.

The article provided a comprehensive overview of the current understanding of the molecular mechanisms by which NRF2 offers cardioprotection and how this may impact cardiovascular health. I also appreciate the balanced discussion of the potential benefits and risks of NRF2 activation in the context of cardiovascular disease and cancer treatment, highlighting the need for further research to fully understand the role of NRF2 in cancer. Therefore, I enthusiastically recommend this manuscript for publication.

Two minor requests for the authors would be

  1. Include a brief discussion regarding the role of senescence as a mechanism for DOX-induced cardiotoxicity and the involvement of NRF2 in this context.

  2. I suggest changing the title since it is very close to a previous review (PMID: 29104732)

Author Response

Response to Reviewer 3 Comments

The authors would like to thank the reviewer for their time reading and assessing the manuscript.

The points presented will be discussed below.

Point 1: Include a brief discussion regarding the role of senescence as a mechanism for DOX-induced cardiotoxicity and the involvement of NRF2 in this context.

Response 1: We note that a discussion about senescence is pertinent within our manuscript given the strong links between NRF2 in ageing and DOX-induced cardiotoxicity. As such, we have included this point of discussion within our review (please see section 6.2) and thank the reviewer for their suggestion.

Point 2: I suggest changing the title since it is very close to a previous review (PMID: 29104732)

Response 2: We accept this comment from Reviewer 3 and have amended the title to be more unique and yet still adequately define the subject of the review.

Round 2

Reviewer 1 Report

The authors did not address most of my comments.